# A Strategy for Identification and Structural Characterization of Compounds from *Plantago asiatica* L. by Liquid Chromatography-Mass Spectrometry Combined with Ion Mobility Spectrometry

**DOI:** 10.3390/molecules27134302

**Published:** 2022-07-04

**Authors:** Hongxue Gao, Zhiqiang Liu, Fengrui Song, Junpeng Xing, Zhong Zheng, Shu Liu

**Affiliations:** 1Jilin Provincial Key Laboratory of Chinese Medicine Chemistry, Changchun Institute of Applied Chemistry, Chinese Academy of Sciences, Changchun 130022, China; mslab34@ciac.ac.cn (H.G.); songfr@ciac.ac.cn (F.S.); xjp@ciac.ac.cn (J.X.); zhengzh@ciac.ac.cn (Z.Z.); 2Institute of Applied Chemistry and Engineering, University of Science and Technology of China, Hefei 230029, China; 3State Key Laboratory of Electroanalytical Chemistry, Changchun Institute of Applied Chemistry, Chinese Academy of Sciences, Changchun 130022, China

**Keywords:** *Plantago asiatica* L., UHPLC-Q-TOF MS, ion mobility spectrometry, macroporous resin

## Abstract

*Plantago asiatica* L. (PAL) as a medicinal and edible plant is rich in chemical compounds, which makes the systematic and comprehensive characterization of its components challenging. In this study, an integrated strategy based on three-dimensional separation including AB-8 macroporous resin column chromatography, ultra-high performance liquid chromatography–quadrupole time-of-flight mass spectrometry (UHPLC-Q-TOF MS), and ultra-high performance liquid chromatography-mass spectrometry with ion-mobility spectrometry (UHPLC-IM-MS) was established and used to separate and identify the structures of compounds from PAL. The extracts of PAL were firstly separated into three parts by AB-8 macroporous resin and further separated and identified by UHPLC-Q-TOF MS and UHPLC-IM-MS, respectively. Additionally, UHPLC-IM-MS was used to identify isomers and coeluting compounds, so that the product ions appearing at the same retention time (RT)can clearly distinguish where the parent ion belongs by their different drift times. UNIFI software was used for data processing and structure identification. A total of 86 compounds, including triterpenes, iridoids, phenylethanoid glycosides, guanidine derivatives, organic acids, and fatty acids, were identified by using MS information and fragment ion information provided by UHPLC-Q-TOF MS and UHPLC-IM-MS. In particular, a pair of isoforms of plantagoside from PAL were detected and identified by UHPLC-IM-MS combined with the theoretical calculation method for the first time. In conclusion, the AB-8 macroporous resin column chromatography can separate the main compounds of PAL and enrich the trace compounds. Combining UHPLC-IM-MS and UHPLC-Q-TOF MS can obtain not only more fragments but also their unique drift times and RT, which is more conducive to the identification of complex systems, especially isomers. This proposed strategy can provide an effective method to separate and identify chemical components, and distinguish isomers in the complex system of traditional Chinese medicine (TCM).

## 1. Introduction

*Plantago asiatica* L. (PAL), belonging to the *Plantago* species, is widely distributed in China and is a traditional Chinese medicine (TCM). Many countries have listed Plantago species as safe herbs in their pharmacopeias. Plantago species have been applied in clinical treatment for centuries [1,2]. PAL has many applications such as antipyretic, diuretic, antitussive, and wound healing properties [3,4] in Asian folk medicine. The chemical composition of PAL has been well reported, and the chemical constituents of PAL are mainly iridoids, triterpenes, phenylpropanoid glycosides, flavonoids, and polysaccharides [5]. A systematic study on the chemical constituents of PAL is helpful to its clinical application. Therefore, extensive research on the chemical constituents of PAL has been conducted. To date, a total of 108 compounds have been identified from PAL. Compounds with significant differences among *Plantago asiatica* L., *P. depressa Willd.,* and *P. major* L. were systematically studied. However, the content of various compounds in PAL varies greatly, and direct analysis may cause trace components to be masked by high-content components. In addition, because there are many similar components and isomers with similar polarity, many chemical components in PAL have not been identified. It is necessary to select suitable methods to enrich these trace components and separate the isomers to identify their structures.

Macroporous resin separation is a common separation method of TCMs [6,7]. The method can exclude high-content components in order to quickly enrich low-abundance components. LC-MS has a good application in the qualitative and quantitative analysis of small molecules from TCMs [8,9,10]. LC can better separate compounds by different polarities and MS can better detect ingredients with low content and acquire their precursor ions and fragment ions to characterize their structures. Its powerful performance has been proposed, and LC-MS showed better resolution and sensitivity [11]. Currently, it is the most popular technique for the analysis of small molecule compounds. However, when using liquid chromatography–mass spectrometry (LC-MS) to detect the chemical components of TCMs, there are co-eluting components and isomers at the same retention time (RT); MS cannot be used alone for the separation and identification of complex chemical components.

Ion mobility–mass spectrometry (IM-MS) was a new technology developed in recent years. IM-MS can separate ionized compounds in a drift tube according to the difference in shape, mass, size, and charge among compounds. [12,13,14]. IMS coupled with UHPLC-MS is appropriate, because a chromatogram is obtained on a minute scale, IMS in milliseconds, and MS in microseconds. In addition, LC and IMS can form a two-dimensional separation system to identify complex compounds. In recent years, more and more studies have been carried out on the identification of small molecules in complex systems by UHPLC-Q-TOF MS combined with IM-MS [15,16]; the combination of multiple detection methods can make the best use of the advantages and avoid weakness. IMS has more applications, especially in distinguishing isomers and co-eluting components [17]. Although the signal strength of MS is higher than that of IM-MS, which causes some fragments to be undetected by IM-MS, LC-MS cannot identify isomers and chromatographically co-eluting compounds. LC-MS detected lots of fragments at the same time, but these fragments did not belong to the same compound, which misled us to identify their fragments. However, because precursor ions and fragment ions of a compound have the same drift time and collision cross-section (CCS) in IM-MS [18], we can distinguish these fragments by which compound they belong to. Therefore, the two methods were combined to identify the compounds together in this study.

In this study, the AB-8 macroporous resin column chromatography was applied to separate and enrich different parts of PAL, and UHPLC-IM-MS and UHPLC-Q-TOF MS were used for the secondary separation and component detection of different parts. Finally, the compounds were identified based on MS information from the chemical database and UNIFI software. Compounds in PAL were identified, the fragmentation mechanism of different kinds of components in MS was studied, and a pair of isomers was analyzed by combining IMS and theoretical calculations. A strategy for isolation, enrichment, analysis, and characterization of compounds was developed, which can be used for the structural analysis and identification of compounds of TCMs.

## 2. Result and Discussions

### 2.1. Application of AB-8 Resin Column Chromatography

TCMs have been widely used and have played an important role in disease treatment. However, the separation and preparation of effective components from TCMs by conventional solvent extraction have many unresolved issues, which has caused the waste of a large amount of raw materials and also high energy consumption [19,20]. Macroporous resins, which can selectively adsorb organics in an aqueous solution owing to their large specific surface, are suitable for separating the compounds of TCMs [21,22]. The employment of macroporous resins for absorbing the ingredients of TCMs can also avoid a lot of waste. There are more than 100 kinds of chemical components of TCMs, some of which are low in content. It is reported that macroporous resins can enrich these minor components by concentrating the solution [7]. Therefore, using macroporous resin column chromatography to separate several types of compounds is very efficient and convenient [23].

It is reported that AB-8 macroporous resin is a weak polar resin and is suitable for separation and purification of weak polar compounds [24]. The compounds of PAL had weak polarity. So, AB-8 macroporous resin was selected in this study. By optimizing the mobile phase and eluent volume, the water and ethanol (15% ethanol, 50% ethanol, and 70% ethanol) were finally selected as the elution phase. Water was used to elute polysaccharides and impurities from the AB-8 macroporous resin column. Figure 1 shows that the main components of 15% ethanol elution fraction (P1) are iridoids and eluted during 0–7 min, the main components of 50% ethanol elution fraction (P2) are phenylethanoid glycosides and eluted during 7–10 min, and main components of 70% ethanol elution fraction (P3) are flavonoids and eluted during 10–24 min. It is especially important that AB-8 macroporous resin can effectively enrich trace compounds such as flavonoids and triterpenoids from PAL. Consequently, the method of combining macroporous resin with UHPLC-Q-TOF MS and UHPLC-IM-MS was useful for analyzing and identifying chemical compounds of PAL.

### 2.2. Characterization of Compounds in PAL

#### 2.2.1. Iridoids

In this study, eleven iridoids from PAL were characterized by their fragments obtained in the negative ion mode (Appendix A). The losses of H_2_O (18 Da), CO_2_ (44 Da), and glucosyl moiety (Glc,162 Da) were detected frequently in iridoids, and a retro-Diels–Alder (RDA) reaction also frequently occurred [25,26]. Take compound 9 as an example to illustrate the identification process of iridoids. The characteristic mass spectra of compound 9 obtained by UPLC-TOF-MS are shown in Appendix A, and the fragmentations pathways of geniposidic acid are shown in Figure 2. Compound 9 was observed at RT 2.05 min, and its precursor ion was observed at *m*/*z* 373. The fragment ions at *m*/*z* 211 [M-Glc-H]^−^ correspond to the loss of the sugar chain from *m*/*z* 373. The [M-Glc-H]^−^ ion further produced [M-Glc-H_2_O-H]^−^, [M-Glc-CO_2_-H]^−^, and [M-Glc-CO_2_-H_2_O-H]^−^ ions at *m*/*z* 193, *m*/*z* 167, and *m*/*z* 149, respectively. The [M-Glc-H]^−^ ion also occurred RDA reaction to produce two fragment ions at *m*/*z* 123 and 86, and another type of RDA reaction occurred to produce an ion at *m*/*z* 101. Finally, compound **9** was identified as a geniposidic acid based on RT and fragment behavior of the reference standard.

The characteristic fragment ions [M-Glc-H]^−^ of compounds **3**, **11,** and **20** were detected at *m*/*z* 183, 213, and 193, respectively. The H_2_O also was the common neutral loss of iridoids, so [M-Glc-H_2_O-H]^−^ ions of three iridoids were also detected at *m*/*z* 165, 195, and 175, respectively. In the negative ion mode, compound **3** was easily combined with HCOOH at *m*/*z* 391. Meanwhile, an RDA reaction occurred to produce ions at *m*/*z* 139; compound **3** was identified as aucubin. As for compound **11**, [M-Glc-CO_2_-H]^−^ and [M-Glc-CO_2_-H_2_O-H]^−^ ions were detected at *m*/*z* 169 and *m*/*z* 151, and compound **11** was identified as 8-epiloganic acid. Compounds **20** showed the two specific losses C_2_H_4_O_2_ (60Da), the losses were detected by glucoside, so [M-C_2_H_4_O_2_-H]^−^ and [M-(C_2_H_4_O_2_)_2_-H]^−^ ions were found at *m*/*z* 295 and *m*/*z* 235. Otherwise, [M-C_3_H_6_O_3_-H]^−^ ion was found at *m*/*z* 265, which was also formed by fragmentation of glycoside. [M-Glc-H]^−^ and [M-Glc-H_2_O-H]^−^ ions were detected at *m*/*z* 193 and 175. Compound **20** was identified as gentiopicroside.

#### 2.2.2. Phenylethanoid Glycosides

Eighteen phenylethanoid glycosides (PhGs) from PAL were characterized in the negative ion mode shown in Appendix A. The neutral loss of phenylethanoid glycosides was 162Da, 152Da, and 146Da, and fragments at *m*/*z* 179 and 161 were detected frequently, which were produced by caffeic acid and anhydroglucose. Compound **42** was determined as acteoside by comparing it with the reference compound. The characteristic mass spectra of acteoside are shown in Figure 3. It shows the [M-Caff-H]^−^ ion at *m*/*z* 461, then the fragment ion at *m*/*z* 315 was obtained by the loss of a rhamnose. The ions at *m*/*z* 179 and 161 were detected as the characteristic ions of phenylethanoid glycosides. Appendix A shows the fragments of acteoside detected by UHPLC-Q-TOF-MS. Compounds **44** and **45** showed the same fragments and *m*/*z* as acteoside, so they are isomers. According to their different RT and references [4], compounds **44** and **45** were determined as isoacteoside and forsythiaside.

The [M-Glc-H]^−^ and [M-Glc-H_2_O-H]^−^ ions detected at *m*/*z* 315 and 298 belong to compound **14**. Anhydroglucose and anhydrophenylethanol were also obtained at *m*/*z* 161 and 135. Therefore, compound **14** was identified as Decaffeoylacteoside.

Compound **49** gave the [M-H]^−^ ion at *m*/*z* 637 and produced the fragment ions at *m*/*z* 461 and 315 after losing 176Da and 146Da in the IM-MS spectrum. We can easily speculate that 176Da was a caffeoyl and a methoxy group; therefore, there is a methoxy group on caffeic acid. Moreover, caffeoyl with a methoxy group at *m*/*z* 193, and the ion at *m*/*z* 175 was detected by loss of an H_2_O (18Da). Meanwhile, the ion at *m*/*z* 443 was obtained by the loss of an H_2_O from the fragment ion at *m*/*z* 461, then the ion was obtained by the loss of a rhamnose. Therefore, Compound **49** was identified as leucoseptoside A.

#### 2.2.3. Flavonoids

Flavonoids are one of the largest classes of natural compounds. The developing study in flavonoids is mainly due to their good antioxidant activity [27]. In this study, twenty flavonoids were characterized from PAL, as shown in Appendix A. The characteristic neutral losses of flavonoids were Glu and H_2_O, and many of these flavonoids have a characteristic fragment at *m*/*z* 151, which is a product of the RDA reaction. Compound **51** (miscanthoside) was taken as an example to illustrate the identification process of flavonoids. Figure 4 shows the possible fragmentation mode of Compound **51.** Compound **51** was detected at *m*/*z* 449. The [M-H-Glc]^−^ ion was detected at *m*/*z* 287, and an RDA reaction occurred to produce ions at *m*/*z* 151 and 135. Therefore, it was determined as miscanthoside. Appendix A shows the fragments of miscanthoside detected by UHPLC-Q-TOF-MS.

Compound **19** showed [M-H]^−^ and [M-H-Glc]^−^ ions at *m*/*z* 481 and 319, then the RDA reaction occurred to produce the fragment ions at *m*/*z* 166 and 151. Compound **19** was identified as ampelopsin glucoside.

Compound **47** generated a precursor ion at *m*/*z* 577. The fragment ion at *m*/*z* 431 was obtained by the loss of a rhamnose, and the fragment ion at *m*/*z* 413 was obtained by the loss of a rhamnose and a H_2_O. Then, the fragment ion at *m*/*z* 269, which lost the glucosyl moiety at the C-7 position, was detected. More fragments were detected with different voltages. It showed two kinds of RDA reactions at *m*/*z* 151 and 146. It was identified as rhoifolin.

#### 2.2.4. Guanidine Derivatives

Guanidine derivatives mainly exist in marine and microbial organisms [19]. Guanidine derivatives can also be used as a disinfectant, which has an inhibitory effect on a variety of bacteria [19,28]. In the positive ion mode, five guanidine derivatives were characterized from PAL, as shown in Appendix A. Guanidine derivatives can peak better in the positive mode because guanidine derivatives are a kind of alkaloid that is easily combined with hydrogen ions to produce a positive charge. The five guanidine derivatives can be detected in the negative ion mode too, but the signal strength is much lower than in the positive ion mode.

Many of these guanidine derivatives have a characteristic fragment at *m*/*z* 84 in the positive ion mode, which is caused by the cleavage of the side chain and the imidazoline ring. The possible fragmentation pathways of compound **30** was shown in Figure 5. The [M-H_2_O-H]^−^ and [M-H_2_O-CO-H]^−^ ions were detected at *m*/*z* 206 and 178, then the [M-H_2_O-CO-H]^−^ ion produced the specific ion of guanidine derivatives at *m*/*z* 84. It lost the branch C_4_H_7_ on the five-membered ring, and the compound can detect the ion at *m*/*z* 168. Compound **30** was identified as plantagoguanidinic acid.

#### 2.2.5. Organic Acids

Organic acids adjust the osmotic pressure and are related to the metabolism of plants, such as organic acids as photosynthetic intermediates [29]. Some organic acids had been widely used in industrial production; for example, citric acid, lactic acid, and succinic acid have been gradually applied in microbial production of organic acids [30].

Three organic acids were determined from the crude extract of PAL shown in Appendix A: gluconic acid (compound **1**), citric acid (compound **6**), and caffeic acid (compound **18**) were determined with the reference compounds. The characteristic neutral losses of organic acids were H_2_O, CO, and CO_2_. For example, citric acid was detected at *m*/*z* 191, and it showed the [M-H_2_O-CO_2_-H]^−^ ion at *m*/*z* 129. The [M-H_2_O-CO_2_-CH_2_-H]^−^ and [M-H_2_O-CO_2_-CO-H]^−^ ions were further observed at *m*/*z* 115 and 101.

#### 2.2.6. Fatty Acids

As the essential molecules for organisms, fatty acids (FAs) consist of long hydrophobic, with carboxylic acid groups at the end [31]. FAs are widely used in food and industry. Meanwhile, FAs also exist in animals and plants. FAs also play an important role in the metabolic process [32].

Eighteen fatty acids were identified from the crude extract of PAL, as shown in Appendix A. Fatty acids contain 16 or 18 carbon atoms. Because the position of the carbon–carbon double bond cannot be determined by MS, their exact structure is uncertain. Take Dihydroxy-stearic acid as an example, [M-H_2_O-H]^−^ and [M-H_2_O-(CH_2_)_9_-H]^−^ ions at *m*/*z* 297 and *m*/*z* 171 were detected, but the specific position of the two hydroxyl groups cannot be determined.

### 2.3. Ion Mobility Separation

#### 2.3.1. The Difference between UHPLC-IM-MS and UHPLC-Q-TOF-MS

UHPLC-IM-MS can detect drift times of compounds by IMS, then the ions were braked to fragments by applying voltage. Therefore, precursor ions and their product ions can be detected at the same drift time, but IMS detected the ions and consumed energy, which reduced the signal strength. For example, leucoseptosideA is a phenylethanoid glycosides, Figure 6A shows the fragments detected by UHPLC-Q-TOF-MS, and Figure 6B shows the fragments detected by UHPLC-IM-MS. UHPLC-Q-TOF-MS can detect all the fragments at the same time. UHPLC-IM-MS also can detect all the fragments; although, some ions have low abundance, it is worth noting that it can distinguish these fragments by their drift times. The MS data were analyzed and processed by UNIFI software, and the fragment ions shown in Figure 6B have the same drift times as the parent ion of leucoseptoside A. This phenomenon can reject fragment ions that do not belong to the parent ion according to the drift times. So, UHPLC-Q-TOF-MS combined with UHPLC-IM-MS was an effective method to apply in identifying compounds of TCMs.

#### 2.3.2. Ion Mobility Separation of a Group of Isomers

Some flavonoids had enol and keto forms, which could be distinguished by IM-MS, NMR, and other instruments [33,34,35]. Because of the different chemical properties between the enol and keto forms, some reports distinguished them by changing the pH, temperature [34], the polarity of the solvent [35], and so on. Plantagoside is a flavonoid in PAL, and previous reports have reported its identification, extraction, and function [36,37,38,39], but no analysis of its isomers has been reported.

In this study, plantagoside and its two isomers were investigated for the first time by IM-MS. Plantagoside compounds **32**, **33**, and **34** are flavonoids. By comparing compound **32** in PAL with the reference compound, the same peaks from the reference compound and PAL in IM-MS were found; they both showed three peaks of different drift times, and drift times of these peaks were 4.16 ms, 4.52 ms, and 5.22 ms, as shown in Figure 7. Their precursor ions were detected at *m*/*z* 465, of which RT at 6.92min was in the negative ion mode. Take compound **32** as an example, it showed the [M-Glc-H]^−^ ion at *m*/*z* 303, then an RDA reaction occurred to produce specific fragments at *m*/*z* 151. The ion at *m*/*z* 313 was detected, which was produced by an RDA reaction. These three compounds have the same RT and fragment ions in LC-MS and cannot be distinguished, but different drift times were detected by IMS. By using Collidoscope software [40], theoretical CCS was calculated to characterize the structure. Compounds **32**, **33**, and **34** were identified as the structures shown in Figure 8. Compounds **32** and **33** were optical isomers. Compound **34** was the enol structure of compound **32** by using theoretical calculations (Table 1). To verify the view, the mobile phase was changed from (A) acetonitrile and (B) water containing 0.1%(*v*/*v*) formic acid to (A) acetonitrile and (B) water containing NH_4_HCO_3_/NH_4_OH (2.5 mmol). It can be seen that compound **34** showed a different abundance between the different mobile phases, and the relative abundance of compounds **32** and **33** had no change. Combining the theoretical calculations and chemical properties of the enol form, compound **34** was identified as the enol form of compound **32**, and both compounds **32** and **33** were keto forms.

## 3. Materials and Methods

### 3.1. Materials and Reagents

PAL was purchased from Hongjian Pharmacy (Changchun, China). PAL was identified by Prof. Qing Huang (Jilin Academy of Traditional Chinese Medicine). HPLC-grade methanol, acetonitrile, and formic acid were obtained from Fisher Scientific (Lough borough, UK); Leucine encephalin and sodium formate were supplied by Waters (Milford, USA). Ammonia solution (NH_3_·H_2_O, purity is 25–28%) and sodium bicarbonate (NaHCO_3_) were purchased from Beijing Chemical Works (Beijing, China).

The standards of acteoside, geniposidic acid, plantagoside, caffeic acid, gluconic acid, citric acid, luteoline, ursolic acid, and oleanolic acid (purity >98%) were acquired from Shanghai Ronghe Bio-Technology Co., Ltd. (Shanghai, China). The chemical structures of reference standards are shown in Figure 9. AB-8 macroporous resin was obtained from Shandong Lukang Record Pharmaceuticals Co., Ltd. Other chemicals were of analytical grade and obtained from Sinopharm Chemical Reagent Co., Ltd.

### 3.2. Preparation of Samples Extract

PAL is the seed, which needs to be broken in traditional Chinese medicine. Therefore, a mortar, a portable traditional Chinese medicine extractor, and an Herbal Blitzkrieg Extractor were, respectively, selected to break the wall of PAL. The results showed that the Herbal Blitzkrieg Extractor had the best wall-breaking effect, so this method was finally selected as the best crushing method. The extraction method was slightly modified by referring to the methods reported in the previous literature [41,42,43]: 20 g PAL was put into 200 mL 75% ethanol solution (*v*/*v*) and extracted by Herbal Blitzkrieg Extractor. The voltage of Herbal Blitzkrieg Extractor was set as 100 V and lasted 100 s. Then, the solution was immersed for 1 h and refluxed for 2 h. Half of the crude extract was concentrated to 1 g/mL. Subsequently, the concentrated solution was adsorbed by AB-8 macroporous resin column (20 mL, 1.6 cm × 10 cm), and eluted with water, 15%, 50%, and 70% ethanol (*v*/*v*) in order at a flow rate of 2 BV/h, the volume of each eluent was 100 mL. Then, the collected three fractions and the remaining crude extract were concentrated and freeze dried for 24 h. Eluted fractions of 15%, 50%, and 70% ethanol are defined as P1, P2, and P3, respectively.

Then, 1 mg of the three separated fractions and crude extract of PAL was weighed and redissolved in 1 mL methanol–water (1:3, *v*/*v*), and ultrasonicated for 30 min. After which, the solutions were filtered through a 0.22 μm membrane filter before being detected by LC-MS.

### 3.3. UHPLC-Q-TOF-MS^E^ and UHPLC-IM-MS^E^ Conditions

The instrument consists of the following parts: a Waters ACQUITY UHPLC system (Waters Corp., Milford, MA, USA) coupled with a Q-TOF SYNAPTG2 High-Definition mass spectrometer (Waters Corp., Manchester, UK) with an electrospray ionization (ESI) source. An ACQUITY UHPLC BEH C18 column (50 mm × 2.1 mm, 1.7 μm, Waters) was used to separate the samples. The mobile phases A and B were acetonitrile and water containing 0.1% (*v*/*v*) formic acid. The column temperature was maintained at 25 °C, and the samples were held at 4 °C. The elution program was as follows: 0–1 min, 5% A isocratic; 1–5 min, 5–15% A linear; 5–7 min, 15–18% A linear; 7–8 min, 18% A isocratic; 8–12 min, 18–48% A linear; 15–17 min, 48–53% A linear; 17–21 min, 53–80% A linear; 21–24 min, 80–100% A linear. Data were obtained both in the negative and positive ion modes. Source parameters for the MS were set as follows: capillary voltage in the positive ion mode was 3.0 kV, and in the negative ion mode was 2.2 kV. Other parameters of the positive ion mode were the same as the negative ion mode: source temperature and desolvation temperature were 100 °C and 400 °C, respectively; cone gas and desolvation gas flow rates were 40 L/h and 500 L/h, respectively. Trap collision energies were 6 V for low collision energy and 25–35 V for high collision energy in UHPLC-Q-TOF MS.

For IM-MS, transfer collision energy was set at 6 V for low collision energy and 15–40 V, 40–50 V, 50–60 V, and 60–70 V for high collision energy, respectively. UHPLC-IM-MS parameters were set as follows: wave velocity and wave height were set at 480 m/s and 28 V, respectively, bias voltage was set to 40 V, and other parameters were the same as UHPLC-Q-TOF MS. UHPLC-IM-MS^E^ data combined with UHPLC-Q-TOF MS^E^ data and were used to identify compounds and found isomers.

### 3.4. Data Analysis

MassLyxn 4.1 software and UNIFI software (Version 14.0, Waters, USA) were used to analyze the MS data. The chemical structural databases, namely, ChemSpider, Pubmed, and Sci Finder, were used for the search and identification of structures of different compounds. UNIFI software was used for processing and visualization of MS data, displaying high and low collision energy spectra, and distinguishing UHPLC-IM-MS^E^ data in high collision energy spectrum by drift times.

## 4. Conclusions

A strategy combining macroporous resin column chromatography, UHPLC-Q-TOF-MS, and UHPLC-IM-MS was proposed to analyze and characterize constitutes from PAL. The strategy involved, macroporous resin column chromatography separating the similar polarity compounds of PAL and enriching the trace compounds. Then, UHPLC-Q-TOF-MS was used to detect the information of the single compounds from PAL, and UHPLC-IM-MS was used to separate the isomers, and the parent ions of the fragment ions with the same RT were identified. Next, MassLyxn 4.1 software, UNIFI software, and Collidoscope software were jointly used to obtain the information from MS. Finally, reference compounds and serval chemical structural databases were used to identify the compounds. In total, 86 components including flavonoids, iridoids, and PhGs were identified by the strategy, and a group of isomers was identified and distinguished by IM-MS and theoretical calculation. The proposed strategy in this study is an effective method for the identification and structural characterization of compounds from PAL.

## Figures and Tables

**Figure 1 molecules-27-04302-f001:**
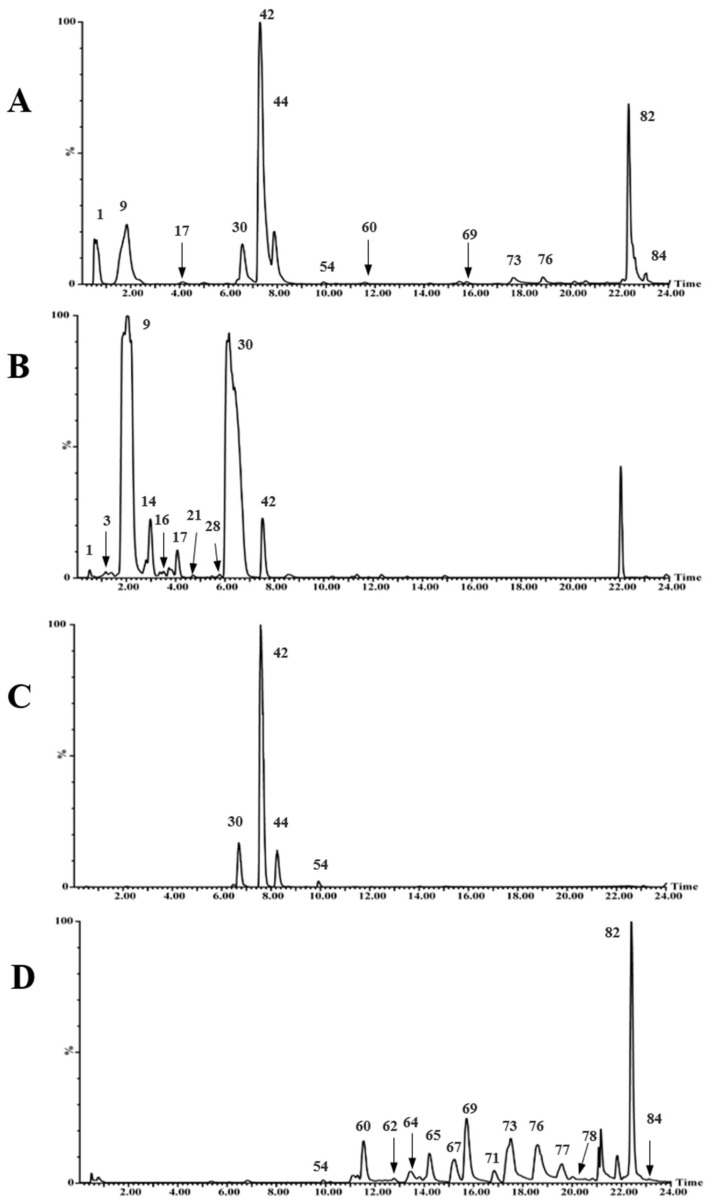
Base peak ion chromatograms of crude extract and each eluted fraction of *PAL* obtained by UHPLC-Q-TOF MS in the negative ion mode: (**A**) crude extract of PAL; (**B**) P1; (**C**) P2; (**D**) P3.

**Figure 2 molecules-27-04302-f002:**
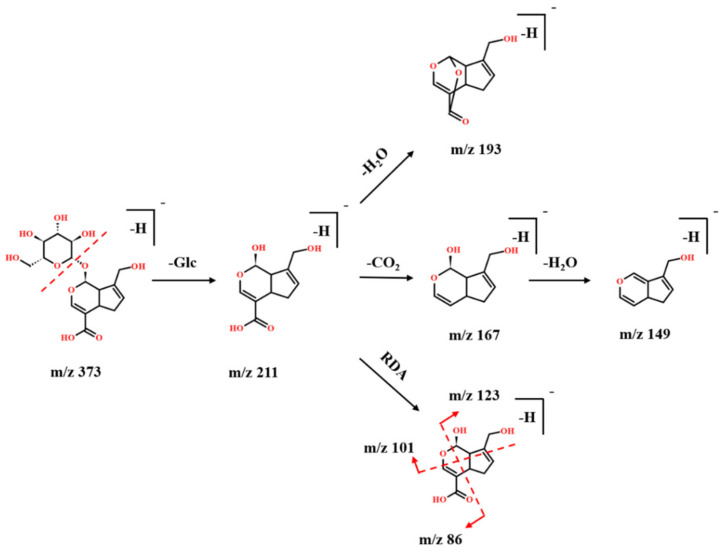
The possible fragmentation pathways of geniposidic acid (Compound **9)** in the negative ion mode.

**Figure 3 molecules-27-04302-f003:**
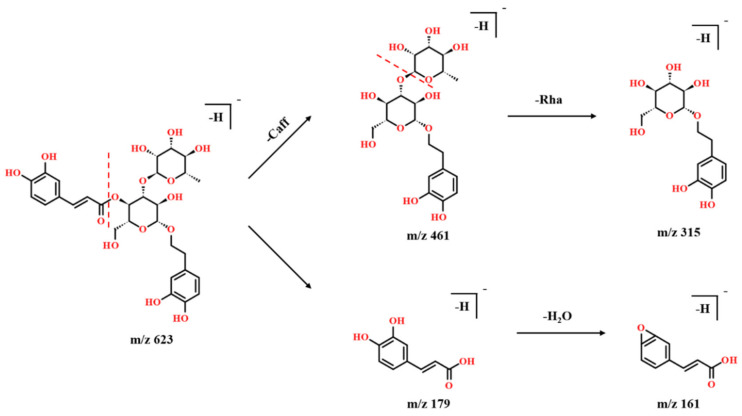
The possible fragmentation pathways of acteoside (compound **42**) in the negative ion mode.

**Figure 4 molecules-27-04302-f004:**
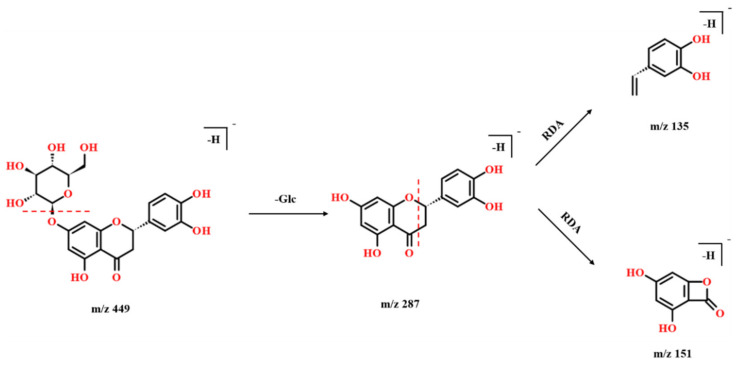
The possible fragmentation pathways of miscanthoside (compound **51)** in the negative ion mode.

**Figure 5 molecules-27-04302-f005:**
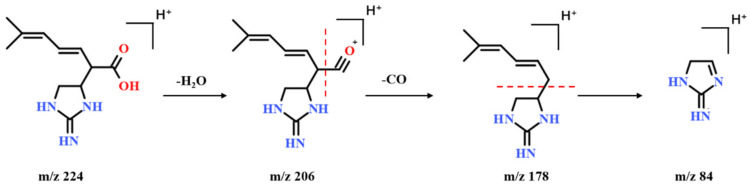
The possible fragmentation pathways of plumbagine B (compound **17**) in the positive ion mode.

**Figure 6 molecules-27-04302-f006:**
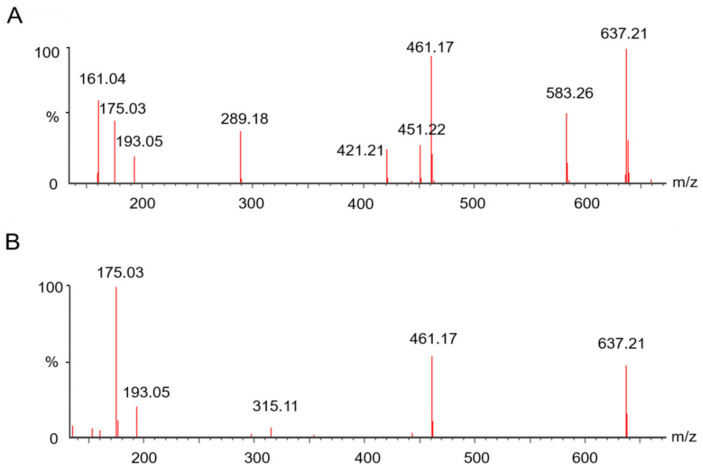
Mass spectrum corresponding to LC peak at 9.16 min: (**A**) the fragment ions were detected by UHPLC-Q-TOF-MS; (**B**) the fragments were detected by UHPLC-IM-MS.

**Figure 7 molecules-27-04302-f007:**
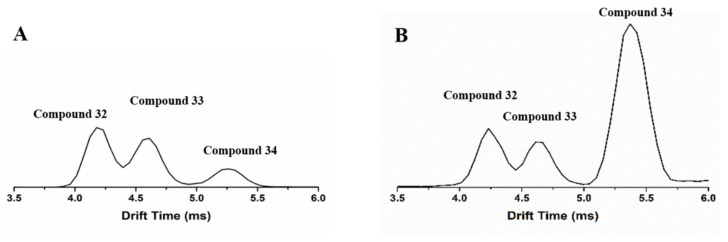
Drift times distribution of compounds **32**, **33,** and **34**: (**A**) the PH of mobile phase is less than 7; (**B**) the PH of mobile phase is greater than 7.

**Figure 8 molecules-27-04302-f008:**
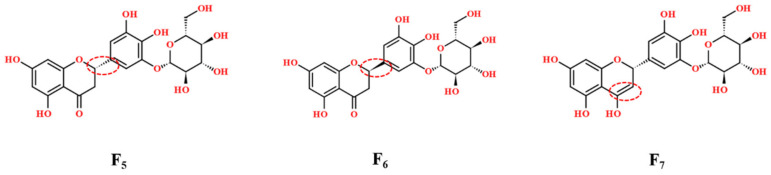
Chemical Structure of compounds **32**, **33**, and **34**.

**Figure 9 molecules-27-04302-f009:**
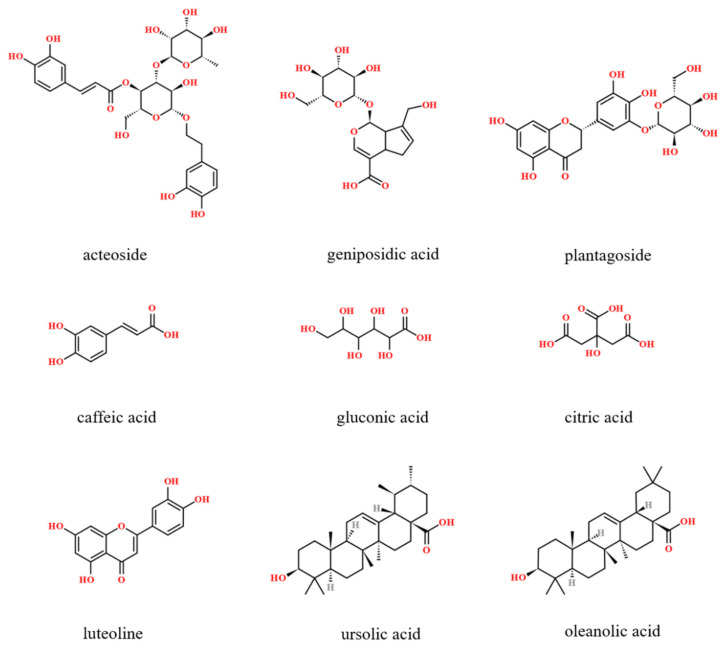
Chemical structures of the reference standards.

**Table 1 molecules-27-04302-t001:** Drift times of compounds **32**, **33,** and **34** by UHPLC-IM-MS and CCS calculated from the drift times.

Compound	Drift time (ms)	CCS/Å^2^
compound **32**	4.16	213.264
compound **33**	4.52	219.395
compound **34**	5.22	237.860

## Data Availability

All data are contained within the article.

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
