# Peer review of "A Strategy for Identification and Structural Characterization of Compounds from Plantago asiatica L. by Liquid Chromatography-Mass Spectrometry Combined with Ion Mobility Spectrometry"

_molecules, 2022, doi:10.3390/molecules27134302_

Round 1
Reviewer 1 Report
Dear authors
At the out set, I would like to congratulate them for the good work.
It would be great if the authors could include the extraction procedure and also reason for the choosing the reported mobile phase in this study. (Have they tried with various mobile phases with different buffers).
Author Response
Thank you for your professional suggestion, we have added relevant descriptions in the revised manuscript. “PAL is the seed, which needs to be broken in traditional Chinese medicine. Therefore, we chose a mortar, a portable traditional Chinese medicine extractor, and an Herbal Blitzkrieg Extractor to break the wall of PAL. The results showed that the Herbal Blitzkrieg Extractor had the best wall-breaking effect, so we chose this method.”
Reviewer 2 Report
Manuscript ID: molecules-1745264
Manuscript entitled “A strategy for identification and structural characterization of compounds from Plantago asiatica L by liquid chromatography-mass spectrometry combined with ion mobility spectrometry” is not novel and not suitable for this journal.
Authors presented work is not novel and presented characterization of compounds in PAL which is not novel as its already studied and published. Also, authors just used different MS techniques to separate the compounds derivatized from PAL using different analytical techniques
Identification and structural characterization of compounds from PAL using analytical techniques is already reported and I don’t see any significance of this work from authors.
Therefore, I recommend rejecting the manuscript.
Author Response
Thank you for your suggestion. In this research, the macroporous resin is used for separation, which not only effectively separates the main chemical substances in PAL by polarity, but also enriches the components with lower content, which is convenient for analysis and identification.
Furthermore, UHPLC-Q-TOF-MS combined with UHPLC-IM-MS was used to carry out extensive component identification, which can effectively identify the parent ions to which the fragment ions belong, and eliminate the interfering fragments, which can save time and effectively identify component (for example Figure 6).
More importantly, a group of isomers was identified and distinguished plantagoside and its isomers by IM-MS and theoretical calculation. To verify the view, this research changed the pH of the mobile phase through the chemical properties of this group of flavonoids, verifying the structures of the compounds as enol form and keto form. The distinction between the isomers of plantagoside of PAL was reported for the first time in this study and listed possible structures.
Reviewer 3 Report
Dear Editors, dear authors,
the manuscript
"A strategy for identification and structural characterization of compounds from Plantago asiatica L by liquid chromatography-mass spectrometry combined with ion mobility spectrometry"
represents an interesting and solidly presented study. After revising it according to my minor comments below, I recommend for publication.
MINOR COMMENTS
Line 41
Please define TCM again
Figure 1, Line 147 and throughout the whole manuscript
Please provide a list in which the compound numbers are assigned to compound names (+ eventually some of their important properties).
Author Response
Q1. Line 41 Please define TCM again
Answer: We have taken the reviewer's suggestion. TCM has been defined in the revised manuscript.
Q2. Figure 1, Line 147 and throughout the whole manuscript
Please provide a list in which the compound numbers are assigned to compound names (+ eventually some of their important properties).
Answer: Compound numbers have been added to the figure legends and throughout the whole manuscript. In addition, in table S1, the number, name, molecular formula and other information of the compound were listed in details.
Reviewer 4 Report
In my opinion, the publication lacks original features, it only shows the possibilities of the ion mobility technique. Language requires strong corrections. I discourage publication of this manuscript.
Author Response
Thank you for your suggestion. In this research, the macroporous resin is used for separation, which not only effectively separates the main chemical substances in PAL by polarity, but also enriches the components with lower content, which is convenient for analysis and identification.
Furthermore, UHPLC-Q-TOF-MS combined with UHPLC-IM-MS was used to carry out extensive component identification, which can effectively identify the parent ions to which the fragment ions belong, and eliminate the interfering fragments, which can save time and effectively identify component (for example Figure 6).
More importantly, a group of isomers was identified and distinguished plantagoside and its isomers by IM-MS and theoretical calculation. To verify the view, this research changed the pH of the mobile phase through the chemical properties of this group of flavonoids, verifying the structures of the compounds as enol form and keto form. The distinction between the isomers of plantagoside of PAL was reported for the first time in this study and listed possible structures.
After your reminder, the English language has also been revised.
Round 2
Reviewer 2 Report
Manuscript ID: Molecules-1745264
Dear Authors,
After revising the manuscript, it's significantly improved and useful in the field. Authors did good job on the supporting information.
Therefore, I accept this paper to published in the Molecules journal.